# OpenReview forum: "Test-Time Fairness and Robustness in Large Language Models"
_ICLR.cc/2025/Conference — Submitted to ICLR 2025_

### Official Review · Reviewer_fuVV · 2024-11-01

**Soundness:** 3
**Presentation:** 3
**Contribution:** 3
**Rating:** 6
**Confidence:** 3

**Summary:**

The paper studies an important problem of ensuring better fairness and robustness of LLM during test time. It focuses on the notion of stratified invariance and advocates the adoption of a stratified data augmentation procedure at test time. The work further implements the procedure on LLMs through prompting (with specially designed role-playing prompts), and naming this strategy out-of-context (OOC) prompting. Extensive empirical validations are done to demonstrate that OOC improved the stratified invariance of LLM predictions and hence fairness in real-world datasets.

**Strengths:**

1. The motivations for the test-time fairness enhancement are well written, and the gaps in the current literature for counterfactual invariance are well discussed.
2. It provides a good analysis of the assumptions, and properly discusses them in the context of practical adoption.
3. The empirical experiments are well-designed to show the superior performance of OOC.

**Weaknesses:**

1. While the authors adequately discussed the assumptions when adopting stratified data augmentation using OCC in the context of LLM, there is no explicit discussion on the limitation of the method in theory and in practice. For example, what are the implications when the assumptions do not hold?

**Questions:**

1. Since the stratifying measurement or the adjustment set $S$ is important to the stratified invariance, can the authors clarify more in the paper how $S$ is typically chosen? I see in the experiments section that $S$ are usually the labels of the task, or empty set.
2. Could the authors clarify what does $S$ being an empty set mean? And how do they affect the context obfuscation/addition steps when it is an empty set?
3. It is assumed that the LLM can incorporate and generate a response containing the new context $z^+$ given the obfuscated input. Do you have empirical results on whether this is indeed the case? And how to test them practically, so that we can determine whether the proposed out-of-context will work with this LLM at test time?
4. It is unclear to me how to read Figure 2. More explanations should be provided.
5. The paper brings causal invariances to LLM inference. While it may be typical for causal invariances to consider classification tasks, it is not how typically LLMs are used in practice. How does the method extend to generative tasks for LLMs?

---

> ### Author Response · Authors · 2024-11-18
> **Rebuttal**
>
> Thank you for your thoughtful feedback, we are glad the reviewer liked our work. We address each of your questions/feedback below.
>
> > While the authors adequately discussed the assumptions... there is no explicit discussion on the limitation of the method...
> >
>
> Our "Practical Considerations" subsection in Sec 3 comprehensively addresses method limitations, including:
>
> - Predicting S;
> - Generating counterfactual augmentations;
> - Requirements for S to be an adjustment set;
> - Computational complexity (newly added, as suggested by another reviewer).
>
> We’d be happy to discuss further if the reviewer has any other possible limitations in mind.
>
> > Can the authors clarify more in the paper how S is typically chosen?
> >
>
> Appendix B provides three concrete examples of choosing S, which directly apply to all tasks in Section 5.1. We've added a reference to this appendix immediately after S's definition for clarity.
>
> > Could the authors clarify what does S being an empty set mean?
> >
>
> An empty $S$ indicates that the user is trying to achieve interventional invariance, i.e. the distribution of the potential outcomes does not change at the population level under different context interventions: $p(y(z)) = p(y(z’)), \forall z,z’$. As we mention in the paper, this is equivalent to requiring that randomized experiments with different contexts have the same outcome distribution.
>
> > Do you have empirical results on whether LLM can incorporate and generate a response containing the new context z^+?
> >
>
> Perfect incorporation of the new context isn’t necessary to achieve stratified invariance, since this variable is sampled independently from the original input. On the other hand, both the obfuscation and the context addition processes could unintentionally remove text that is important for the prediction of the task, impacting the method’s predictive performance —which we observe in Figure 3 to not be the case.
>
> > It is unclear to me how to read Figure 2.
> >
>
> Figure 2 shows bias reduction effectiveness:
>
> - X-axis: Difference between default prompting SI-bias and each method's SI-bias.
> - Positive values (right of dashed line): Bias reduction.
> - Negative values (left of dashed line): Bias increase.
> - Bar length: Magnitude of change.
>
> We expanded the caption for clarity, thanks for this feedback!
>
> > How does the method extend to generative tasks for LLMs?
> >
>
> Our method can be applied to open-ended generation tasks. The only requirements are the user ability to define/describe S, Z and sample from Z. There are no requirements on the output $Y$ that restricts open-ended generation.
>
> We welcome any follow-up questions or need for clarification.

---

> > ### Comment · Reviewer_fuVV · 2024-12-02
> >
> > I would like to thank the authors for the response. I do not have further questions and I am happy with the paper. I will keep my positive opinion about the work.

---

> ### Author Response · Authors · 2024-11-23
> **Discussion**
>
> Dear reviewer, we look forward to your comments. Please, let us know if there's anything left for us to address.

---

> ### Author Response · Authors · 2024-11-25
> **Discussion ends soon**
>
> As the discussion period ends tomorrow, we are looking forward to your comments and feedback on our rebuttal. Thank you again for your service.

---

> > ### Author Response · Authors · 2024-12-01
> > **Reminder**
> >
> > Dear reviewer, discussion ends tomorrow, and we hope we can address any further questions you might have about our work.

---

### Official Review · Reviewer_cCsx · 2024-11-02

**Soundness:** 3
**Presentation:** 2
**Contribution:** 2
**Rating:** 6
**Confidence:** 2

**Summary:**

This paper considers the problem of LLM debiasing at the test time. By making use of causal invariance, the authors proposed a novel stratified invariance notion to address the limitation of standard conterfactual data augmentation. Besides, an out-of-context prompting strategy, inspired by stratified invariance, was proposed to demonstrate that the bias of LLMs can be reduced for real-world benchmarks.

**Strengths:**

- Originality:
    - Unlike previous works that used safety instructions to implicitly address the bias issue, this work leverages the causal invariance framework that utilizes interventions to obtain a less biased result.
    - This work also developed a stratified invariance notion that is built on observational data (random generations).
    - A novel OOC strategy is introduced to debias LLM predictions.

- Quality:
    - The theoretical definition and analysis are introduced for stratified invariance which makes the design of OOC in principle.
- Clarity:
    - The clarity could be improved.
- Significance:
    - The proposed method evaluated on stratified invariance bias shows significant improvement, but it is unclear on other evaluation metrics.

**Weaknesses:**

- The presentation of this paper could be substantially improved. I tried very hard to understand this paper, but many things still remain unclear. I will list a few here:
    - Line 127-138 are helpful for understanding but they only appeared in the method section. I suggest the author can elaborate the problem and objective further in the introduction.
    - Motivation of applying causal invariance in LLM debiasing is unclear.
    - It would be better if in Sec. 3 or before, a complete example in LLMs where the introduced variable can have some correspondence. I can only find such correspondence in Sec. 5.1.
    - Notation abusing makes some concepts confusing: e.g., what does "a.s." & "d" over = mean? in distribution?;
    - typos: e.g., "predictins" in line 215
- Why does OOC perform less significant on LLAMA-3-70B? Is it because the possible obfuscation instructions are of low quality?
- To measure if your method successfully remove/reduce the bias, except for the stratified invariance you introduced, are there other bias evaluation metrics that you can use to demonstrate?

**Questions:**

See questions above.

---

> ### Author Response · Authors · 2024-11-18
> **Rebuttal**
>
> Thank you for your review. We’ll address your questions and suggestions next.
>
> > Line 127-138 are helpful for understanding but they only appeared in the method section. I suggest the author can elaborate the problem and objective further in the introduction.
> >
>
> We’ve updated the introduction by distilling this paragraph. Please, let us know if this addresses your concerns around the motivation and intuition.
>
> > It would be better if in Sec. 3 or before, a complete example in LLMs where the introduced variable can have some correspondence. I can only find such correspondence in Sec. 5.1.
> >
>
> We have added a sentence in Sec. 3 referring to Fig. 1 as an example.
>
> > Notation abusing makes some concepts confusing: e.g., what does "a.s." & "d" over = mean? in distribution?;
> >
>
>  $\overset{a.s.}{=}$ and $\overset{d}{=}$ are standard notation that stand for almost sure equality and equality in distribution, respectively.
>
> > Typos: e.g., "predictins" in line 215
> >
>
> Fixed. Please let us know if you notice any other typos.
>
> > Why does OOC perform less significant on LLAMA-3-70B? Is it because the possible obfuscation instructions are of low quality?
> >
>
> OOC's performance on LLAMA-3-70B is actually comparable to GPT:
>
> - GPT: Bias reduction in 8/9 tasks.
> - LLAMA: Bias reduction in 7/9 tasks.
>
> The main difference appears in the clinical notes dataset, where:
>
> - Inputs are significantly longer than other datasets.
> - LLAMA seems to show difficulty with obfuscation on these longer inputs.
> - Detailed results are available in Appendix E for better reference.
>
> > To measure if your method successfully remove/reduce the bias, except for the stratified invariance you introduced, are there other bias evaluation metrics that you can use to demonstrate?
> >
>
> Variants of our SI-bias metric have appeared in fairness literature. For simplicity, let’s consider our metric for binary variables $\hat{Y},Z$:
>
> 1. **Conditional Independence:**
>     - $\hat{Y} \perp Z \mid S$ holds if and only if:
>     - $P(\hat{Y}=1 \mid Z=1, S=s) - P(\hat{Y}=1 \mid Z=0, S=s) = 0$ for all $s \in \mathcal{S}$.
> 2. **Our Metric Choice:**
> - $gap(s) = \mid  P(\hat{Y}=1 \mid Z=1, S=s) - P(\hat{Y}=1 \mid Z=0, S=s) \mid$ .
> - SI-bias measures maximum gap(s) across all s values.
> - Preferred over averaging to avoid masking group-specific biases.
> 1. **Validation:**
>     - Supported by classical work [1].
>     - Used in modern causal approaches [2].
>     - Any metric quantifying conditional independence would capture the same property.
>
> We're happy to discuss specific alternative metrics if you have any in mind.
>
> [1] Hardt, Moritz, Eric Price, and Nati Srebro. "Equality of opportunity in supervised learning." Advances in neural information processing systems 29 (2016).
>
> [2] Veitch, Victor, et al. "Counterfactual Invariance to Spurious Correlations: Why and How to Pass Stress Tests.”

---

> ### Author Response · Authors · 2024-11-23
> **Discussion**
>
> Dear reviewer, we look forward to your comments. Please, let us know if there's anything left for us to address.

---

> ### Author Response · Authors · 2024-11-25
> **Discussion ends soon**
>
> As the discussion period ends tomorrow, we are looking forward to your comments and feedback on our rebuttal. Thank you again for your service.

---

> > ### Comment · Reviewer_cCsx · 2024-12-01
> > **Thanks for the response**
> >
> > Dear authors,
> >
> > Sorry for the late update. The introduction looks a bit clearer now, but I still suggest the authors add more motivation or justification for choosing a causal inference lens. Notation-wise, I suggest the authors could use subscripts for some abused variables. For example, the authors used $\hat{Y} := \hat{Y}(Z)$ but sometimes directly used $\hat{Y}$ as a general variable, where the dependence looks confusing. This also applied to $X$ and $Z$.
> >
> > Thanks for your effort and clarifications. They helped address some of my concerns, so I raised my rating accordingly.

---

> > > ### Author Response · Authors · 2024-12-01
> > > **Thank you**
> > >
> > > Thank you for the feedback and raising your score to acceptance. Just as a final clarification, the definition $\hat{Y} := \hat{Y}(Z)$ is important since it connects the observed label $\hat{Y}$ to the intervened labels $\hat{Y}(z), z \in \mathcal{Z}$. One equivalent way of defining $\hat{Y}$ is $\hat{Y}:= \sum_{z \in \mathcal{Z}} \hat{Y}(z) \cdot \mathbf{1}( Z=z)$. We agree there's a slight abuse of notation in the former, so we will add the latter to the camera-ready version. Thank you once again for getting back to us and your service.

---

### Official Review · Reviewer_86wE · 2024-11-04

**Soundness:** 3
**Presentation:** 3
**Contribution:** 3
**Rating:** 6
**Confidence:** 4

**Summary:**

The paper considers the test-time evaluation of fairness for LLMs. In particular, the paper aims to address the potential issue of naive applications of certain causal debiasing strategies (e.g., in terms of counterfactual data augmentations operating on the individual level), and proposes Stratified Invariance (Definition 1). The idea is to incorporate additional measurements at test time, so that stratified predictors can be constructed (for bias evaluation purposes). Prompting template and empirical results are presented.

**Strengths:**

The strength of the paper comes from the clear presentation of the potential issue of directly applying certain causal fairness notions (especially ones that are related to counterfactual invariance) in the LLM context (Section 2), and the attempt to address this issue by proposing stratified invariance (Definition 1), which is a reasonable middle ground between the almost-sure-equality between potential outcomes (counterfactual invariance) and the distribution-level equality (referred to as intervention invariance in the paper). The theoretical presentation (Section 2) is relatively clear and not hard to follow, the OOC prompting design (Section 3) is guided by the theoretical analysis, and the empirical evaluations include how OOC improve stratified invariance, as well as how to approach counterfactual invariance through stratifications.

**Weaknesses:**

The paper can be improved by (1) considering recent LLM debiasing strategies that do not specifically "rely on model's implicit understanding of bias" (lines 47 -- 49), so that the addressing of the existing LLM literature can be more comprehensive; (2) including discussion on the inference overhead of the proposed pipeline.

(1) recent LLM debiasing strategies that do not specifically rely on model's implicit understanding of bias

The paper presents criticisms of the existing LLM debiasing strategies in terms of the reliance on model's implicit understanding of bias (lines 47 -- 49). However, this might not be the case for recent LLM debiasing strategies. For instance, Li et al. (2024) consider a possible causal modeling of how LLM decisions are modulated by prompts, and proposed prompting-based strategies to encourage fact-based reasoning where no social category (e.g., gender, race) appears.  These strategies do not rely on model's understanding of bias. Considering them would help make the addressing of existing very relevant literature more comprehensive.

(2) discussion on the inference overhead of the proposed pipeline

Since the proposed approach (OOC prompting) involves multiple inference followed by a majority vote, it seems that the inference cost goes up pretty quickly. It would be important to discuss the relation between the inference overhead introduced by the proposed approach and the effectiveness of debiasing.

#### Reference

Li, J., Tang, Z., Liu, X., Spirtes, P., Zhang, K., Leqi, L., & Liu, Y. (2024). Steering LLMs Towards Unbiased Responses: A Causality-Guided Debiasing Framework. arXiv preprint arXiv:2403.08743.

**Questions:**

As detailed in comments in the section above.

---

> ### Author Response · Authors · 2024-11-18
> **Rebuttal**
>
> Thank you for your thoughtful feedback and positive comments. We address both of your suggestions below.
>
> > The paper presents criticisms of the existing LLM debiasing strategies... However, this might not be the case for recent LLM debiasing strategies. For instance, Li et al. (2024)...
> >
>
> We have thoroughly analyzed Li et al.'s FACT approach in our general response, including new comparative experiments that clarify the key differences between our methods and their limitations.
>
> > Since the proposed approach (OOC prompting) involves multiple inference followed by a majority vote, it seems that the inference cost goes up pretty quickly...
> >
>
> We have added a computational complexity analysis to the practical considerations part of Sec 3:
>
> **Complexity Analysis:**
>
> - Original inference: $O(N_I^2 + N_I \times N_O)$, where $N_I$ := input size, $N_O$ := output size.
> - OOC (single prediction): $O(5N_I^2 + N_I \times N_O)$.
> - OOC with variance reduction: $O(m \times (5N_I^2 + N_I \times N_O))$, where $m$ := number of repetitions.
>
> Importantly, our experiments show that small values of $m$ (1 or 3) are sufficient for robust results, keeping the computational overhead manageable and constant relative to input size.
>
> We welcome any additional questions or comments about our work.

---

> ### Author Response · Authors · 2024-11-23
> **Discussion**
>
> Dear reviewer, we look forward to your comments. Please, let us know if there's anything left for us to address.

---

> ### Author Response · Authors · 2024-11-25
> **Discussion period ends soon**
>
> As the discussion period ends tomorrow, we are looking forward to your comments and feedback on our rebuttal. Thank you again for your service.

---

> > ### Author Response · Authors · 2024-11-29
> > **New experiment feedback**
> >
> > The reviewer pointed two weaknesses of the paper, which we believe we have addressed in the rebuttal submitted 10 days ago. Could you please clarify whether your concerns were addressed? In case we have time to provide further clarifications.

---

> ### Comment · Reviewer_86wE · 2024-11-30
> **Thank Author(s) for the Responses**
>
> Thank authors for the responses. The clarifications and further discussions address the original concerns to a certain extent. I do not have further questions from my end, but I would encourage authors to consider presenting the connections to, and differences from, very related previous works in a more comprehensive and transparent way. When highlighting the strength of the proposed approach, correctly acknowledging the scope/setting where previous works are able to handle may also be necessary and important. For instance, to the best of my knowledge, Li et al. (2024) can handle scenarios beyond unfairness induced by selection bias, and the role of selection mechanism goes beyond counteracting existing selection bias.
>
> I will keep my evaluation at the positive side.

---

> > ### Author Response · Authors · 2024-11-30
> > **Thank you for the feedback**
> >
> > We thank the reviewer for getting back to us and for remaining positive about our work. We agree it's important to address related work, which we believe we did in the revised draft. There, we never refer to FACT as a work specifically about fairness, but rather selection bias, which is a concept usually defined using selection mechanisms, see [1] for an example. The reviewer mentions we have _"addressed the concerns to some extent"_. Could you please provide us with concrete changes you would like to see in the final version? We would be more than happy to hear and improve the work.
> >
> > [1] https://proceedings.mlr.press/v22/bareinboim12.html

---

### Official Review · Reviewer_Q4ph · 2024-11-05

**Soundness:** 2
**Presentation:** 1
**Contribution:** 2
**Rating:** 5
**Confidence:** 5

**Summary:**

This paper proposed stratified invariance, a stratified notion of debiasing, to capture a range of debiasing requirements from population level to individual level through an additional measurement that stratifies the predictions. The authors further propose Out-Of-Context (OOC) prompting, a zero-shot method that simulates stratified counterfactual data augmentations in LLM predictions.

**Strengths:**

The theoretical development of Stratified Invariance and Stratified Data Augmentation is interesting. Also, by experimenting on both synthetic and real-world datasets, the authors demonstrate the advantage of the proposed prompting strategy to boost stratified invariance in LLM predictions at test time.

**Weaknesses:**

While the paper’s introduction of "stratified invariance" is an interesting measure of fairness, it appears conceptually close to existing techniques in fair representation learning and causal fairness (e.g., statistical parity). It would be good if the authors could provide an in-depth discussion with other fairness metrics or write out the equations for comparison if this measurement is claimed as a novelty. It is also worth noting that the proposed metric and/or prompting strategy only works for decision tasks (i.e., the output is a discrete answer/decision). The paper also lacks discussions with other debiasing methods, particularly those leveraging causal inference, such as the causality-guided debiasing framework proposed by Li et al. (2024) and Si et al. (2023). The OOC prompting strategy is also very similar to the FACT strategy in Li et al. (2024): i.e., the transformation of the examples in Figure 1 is very similar to Figure 1 in  Li et al. (2024). It would also be nice to conduct comparative analyses with these existing prompting approaches to contextualize the contributions and highlight relative advantages. I am willing to increase my score if the authors could address these concerns.

- Li, Jingling, et al. "Steering LLMs Towards Unbiased Responses: A Causality-Guided Debiasing Framework." arXiv preprint arXiv:2403.08743 (2024).
- Si, Chenglei, et al. "Prompting gpt-3 to be reliable." arXiv preprint arXiv:2210.09150 (2022).

**Questions:**

1. Could the authors provide concrete examples to better illustrate Definition 2 and Definition 3? Specific examples could clarify these definitions and their practical implications, helping readers understand the core concepts more intuitively. A worked example applying these definitions to a simple fairness scenario would help illustrate how they capture different aspects of fairness.

2. Could the authors offer intuitive justifications for Lemma 1 and Theorem 1? An explanation or reasoning beyond formal proofs would help make these results more accessible and easier to interpret.

3. Could the authors conduct comparative analyses with existing prompting approaches to contextualize the contributions and highlight relative advantages (see weakness section)?

4. Could the authors provide an in-depth discussion on how this work aligns with or differs from other fairness metrics?

---

> ### Author Response · Authors · 2024-11-18
> **Rebuttal**
>
> Thank you for your thoughtful feedback. We address each point below.
>
> > While the paper's introduction of "stratified invariance" is an interesting measure of fairness, it appears conceptually close to existing techniques... It would be good if the authors could provide an in-depth discussion with other fairness metrics...
> >
>
> Our related work section (”other fairness metrics”) explains how stratified invariance connects to established fairness concepts:
>
> 1. **Relationship to Traditional Metrics:**
>     - Demographic parity ($\hat{Y} \perp Z$).
>     - Equalized odds ($\hat{Y} \perp Z \mid Y$).
>
>     Per Lemma 1, these are special cases of stratified invariance where $S$ is either empty or contains only the label.
>
> 2. **Causal Fairness Connection:**
> While counterfactual invariance is prominent in causal fairness literature, our "Counterfactual Invariance" section demonstrates that several works attempting counterfactual invariance actually achieve stratified invariance.
>
> We have expanded this discussion in the manuscript.
>
> > ...the proposed metric and/or prompting strategy only works for decision tasks...
> >
>
> The discrete variable assumptions are already required by current language models that also model discrete variables, i.e. only model discrete distributions.
>
> > The OOC prompting strategy is very similar to the FACT strategy in Li et al. (2024)...
> >
>
> Please see our detailed comparison in the general response, which demonstrates key differences in assumptions and empirical performance comparison.
>
> > Could the authors provide concrete examples to better illustrate Definition 2 and Definition 3?
> >
> - **Definition 2 (Adjustment Sets):** We provide three concrete examples in Appendix B (now referenced directly after Definition 2).
> - **Definition 3 (OOC Algorithm):** Figure 1 provides a complete example of OOC in action.
>
> > Could the authors offer intuitive justifications for Lemma 1 and Theorem 1?
> >
>
> **Lemma 1:**
>
> - Challenge: Testing causal properties with only observational data.
> - Key result: Lemma 1 shows that with an adjustment set S, observing $(Z,S,\hat{Y})$ is sufficient, and testing for stratified invariance reduces to checking the conditional independence $\hat{Y} \perp Z \mid S$.
> - Intuition: $S$ being an adjustment set allows us to test the potential outcomes $\hat{Y}(z)$ through the observed $Y$, and the conditional independence assumption guarantees the invariance of the conditionals in Definition 1.
>
> **Theorem 1 Intuition:**
>
> - Challenge: Pure counterfactual invariance is unattainable at test time even with a counterfactual data augmentation machine.
> - Key result: Counterfactual augmentations of input $X$ and $S$ can achieve stratified invariance.
> - Intuition: When we generate a counterfactual augmentation, we have to resample any randomness not in $S$ used to generate the input, and this may not coincide with the one that generate the observed input. This is essentially the gap between stratified and counterfactual invariance, so as $S$ contains more of the randomness in $X$, we approach counterfactual invariance.
>
> We welcome any follow-up questions or requests for further clarification.

---

> ### Author Response · Authors · 2024-11-23
> **Discussion**
>
> Dear reviewer, we look forward to your comments. Please, let us know if there's anything left for us to address.

---

> ### Author Response · Authors · 2024-11-25
> **End of discussion period is approaching.**
>
> As the discussion period ends tomorrow, we are looking forward to your comments and feedback on our rebuttal. Thank you again for your service.

---

> > ### Author Response · Authors · 2024-11-29
> > **Last days for Clarification**
> >
> > Dear reviewer,
> >
> > You explicitly mentioned in your review:
> >
> > > I am willing to increase my score if the authors could address these concerns.
> >
> > Could you please clarify whether we have not addressed your concerns?

---

> > > ### Author Response · Authors · 2024-12-01
> > > **Did we address your concerns?**
> > >
> > > Dear reviewer, tomorrow is the last day of discussion. We believe we have addressed the questions and request for additional experiments in your initial review. Could you please clarify that?

---

> ### Comment · Reviewer_Q4ph · 2024-12-01
> **Still Not Yet Ready for Publication: Constructive Feedback I**
>
> First, I would like to thank the authors for their response and for the effort they have put into addressing the comments provided. I also want to acknowledge that it took me a considerable amount of time to make all my suggestions as constructive as possible, as I genuinely want to help improve this work rather than just criticize it.
>
> After carefully reading through the updated manuscript and all rebuttals (including ones to other reviewers), I must say that the revised submission still falls short of the quality and rigor required for publication at ICLR. While I appreciate the time and effort the authors have invested in this work, I must emphasize that the decision here must be made based on merit rather than effort. My concerns, some of which have intensified after reading the revised version and responses, are outlined below. I also think it would be fairer and more beneficial/impactful to the ML and fairness community if this work could address the following key aspects:
>
> ### I. New concerns come out with the current comparison with the work from Li et al. (2024):
> 1. **Unfair Experimental comparison**: Li et al. (2024) proposed a comprehensive causal framework rather than a single strategy to address potential biases in LLM decision-making. Notably, the Fact strategy they presented only regulates the bias flow on a subset of all causal pathways that could lead to objectionable dependence between the LLM's output on the demographic information (i.e., protected/spurious attributes, as used by the authors). That's why Li et al. (2024) proposed three distinct prompting strategies to debias the LLM's outputs altogether so that all causal pathways can be regulated to some extent (more details can be found in Figures 4 and 7 in [1]).
> 2. **Missing attribution**: There appears to be a close conceptual alignment between context obfuscation/addition in section 3 of your work and the 'demographic-agnostic fact'/'demographic-aware text' mentioned in [1]. More specifically, context obfuscation and context addition seem to correspond to the processes or implementations used to derive the 'demographic-agnostic fact' and 'demographic-aware text'. This conceptual alignment should be explicitly acknowledged to provide proper context and attribution. At the same time, such realizations are nontrivial, and the well-designed role-play prompts should be considered key novelties of this work.
> 3. **The current implementation of OOC is essentially applying a selection mechanism to debias model's output**: The current implementation of the OOC prompting strategy could be framed more explicitly as a selection mechanism that regulates specific causal pathways in the LLM’s decision-making process. As outlined in steps 4-6 of Algorithm 1, the process generates a new context addition for each sampled protected attribute and then gathers the LLM's answer to each generated context addition. This process effectively applies a selection mechanism over the node "demographic-aware text representation" (i.e., $X^+_{LM,j}$) and the node "demographic representation" (i.e., $Z^+_j$) in the causal diagram (referenced in Figure 4c of [1]). This mechanism regulates the biased information flow along edge 1, but it does not address other potential causal pathways through the "demographic-agnostic text representation" that could inject bias into LLM's potential decision. As detailed in II.2 below, this could result in residual bias or complete failure under certain circumstances. Recognizing these limitations while positioning OOC as a distinct strategy within the broader causal framework proposed by Li et al. (2024) could further enrich the discussion.
>
> **Suggestion**: The authors could acknowledge the conceptual alignment between context obfuscation/addition and 'demographic-agnostic fact'/'demographic-aware text' from [1]. At the same time, you can emphasize the nontrivial effort required to systematically derive these texts and highlight the innovative use of role-play prompts to guide the LLM in generating such texts automatically.
>
> Moreover, the OOC prompting strategy could be framed as a novel individual strategy within the causal framework proposed in [1]. If treated as an individual strategy, it would be reasonable to just compare it directly with the FACT strategy from [1]. Still, it's worth mentioning the value of combining multiple strategies (addressing different causal pathways) to achieve better overall performance.
>
> [1] Li, J., Tang, Z., Liu, X., Spirtes, P., Zhang, K., Leqi, L., & Liu, Y. (2024). Steering LLMs Towards Unbiased Responses: A Causality-Guided Debiasing Framework. arXiv preprint arXiv:2403.08743.

---

> ### Comment · Reviewer_Q4ph · 2024-12-01
> **Still Not Yet Ready for Publication: Constructive Feedback II**
>
> ### II. Key Assumptions Underpinning the Success of OOC are missing:
> While the theoretical definitions and assumptions in this work are self-contained, the actual implementation of OOC needs to assume the following conditions—which are not addressed in the current manuscript—to work as expected.
> 1. **Uniform Sampling of the protected or spurious characteristic**
>
> Step 4 in Algorithm 1 needs uniform sampling of the protected attributes $Z_j^+$ from the set $\mathcal{Z}$. This implies the prior knowledge of the complete set $\mathcal{Z}$. While this assumption may be feasible with benchmark datasets where all unique values of $\mathcal{Z}$ can be derived from the data, in real-world scenarios, this may be impractical. For example, $\mathcal{Z}$ could represent complex attributes like physical appearances (e.g., in BBQ dataset [2]), and all physical appearances could encompass an undefined or large set.
>
> **Suggestion**: The authors should explicitly state this assumption when describing OOC or Algorithm 1 and discuss its implications. Uniform sampling from a known, predefined set is achievable; however, as the scope of protected attributes grows (e.g., to encompass diverse physical appearances or other complex characteristics), new methods to define and sample from $\mathcal{Z}$ could be a valuable avenue for future research.
>
>
> 2. **Assumption of Unbiased Internal Model Knowledge for Majority**:
>
> The effectiveness of OOC’s majority voting step (Step 8 in Algorithm 1) assumes that the model's internal knowledge is unbiased. If the model exhibits systemic biases due to its training data—for instance, associating specific outcomes with certain demographic groups—the majority voting mechanism may reinforce these biases instead of mitigating them. Specifically, By doing majority voting, OOC essentially is doing an aggregation of the LLM's answers to all added context $X^+_{LM}(Z)$ for all $Z \in \mathcal{Z}$.
>
> For instance, if a model consistently associates a positive outcome with a specific demographic group due to historical biases in the training data: say that the LLM will only output "No" to the following input "given ... scenario, decide whether {a given ehnicity} is a criminal" when the given ethnicity is assigned to be White, and the LLM will output "Yes" to all other ethnicities. When the ground truth on the given scenario is indeed "No", in this case, OOC prompting will fail to correct this historical bias as it will output "Yes" for all ethnicities based on majority voting. The ideal debiasing outcome is to answer correctly for all demographic groups rather than answer incorrectly for all groups.
> The above is also a failure case discovered by Li et al. (2024) (details in their Table 2): while the model may answer the base/fact question incorrectly, it could answer the original question correctly only for a particular demographic group. As addressed in the above section I.3, the OOC Strategy essentially applies a selection mechanism over the node "demographic-aware text representation" (i.e., $X^+_{LM,j}$) and the node "demographic representation" (i.e., $Z^+_j$) in the causal diagram (referenced in Figure 4c of [1]). This mechanism regulates the biased information flow along edge 1, but it does not address other potential causal pathways through the "demographic-agnostic text representation" that could inject bias into LLM's potential decision. Therefore, implementing other selection mechanisms to counteract such historical biases (i.e., Strategy II and III in [1]) is crucial to ensure fair and accurate outcomes across all demographic groups.
>
> **Suggestion**: To bridge the gap between the theoretical claims of the manuscript and the practical implementation of OOC, the authors should explicitly acknowledge this assumption and its implications. Adding this assumption would clarify that the effectiveness of OOC relies on the model’s underlying representations being relatively unbiased. Furthermore, the authors could mention the importance of applying additional selection mechanisms to address such historical biases intrinsic to the pretrained models.
>
> [1] Li, J., Tang, Z., Liu, X., Spirtes, P., Zhang, K., Leqi, L., & Liu, Y. (2024). Steering LLMs Towards Unbiased Responses: A Causality-Guided Debiasing Framework. arXiv preprint arXiv:2403.08743.
> [2] Parrish, A., Chen, A., Nangia, N., Padmakumar, V., Phang, J., Thompson, J., ... & Bowman, S. R. (2021). BBQ: A hand-built bias benchmark for question answering. arXiv preprint arXiv:2110.08193.

---

> ### Comment · Reviewer_Q4ph · 2024-12-01
> **Still Not Yet Ready for Publication: Constructive Feedback III**
>
> ### III. The Current Presentation of the Work Still Has Significant Room for Improvement
> 1. **Mapping Out-of-Context (OOC) Prompting to Stratified Data Augmentation**
>
> The authors claim that OOC is a prompting strategy to **implement** stratified data augmentation. However, the current demonstration of OOC in Figure 1c does not convincingly align with Definition 3 (Stratified Data Augmentation). The term implementation implies a strong and direct realization of stratified data augmentation, which currently lacks sufficient justification. Specifically, the demonstration in Figure 1c does not adequately support the claim that OOC is indeed an implementation.
>
> **Suggestion**: To substantiate this assertion, additional justifications or clearer illustrations are necessary. For instance, it may be clearer to have a consistent example throughout Figure 1c, Definition 3, and Algorithm 1.
>
> 2. **Improving the Illustration of Algorithm 1 with Real-World Contexts**
>
> The explanation of Algorithm 1 would greatly benefit from the inclusion of a real-world example that demonstrates its application to a specific dataset. Such an illustration would also make the algorithm more accessible and comprehensible to a broader audience.
>
> 3. **Detailed Points for Consideration**
>
> How is the parameter $m$ chosen in Algorithm 1? A detailed explanation or a discussion of guidelines for this selection could be helpful.
> As suggested by Reviewer `cCsx`, the manuscript would benefit from additional definitions where necessary to ensure that key terms and concepts are explicitly clear to readers.
>
> ---
>
> In light of the issues outlined above, I feel that the current manuscript is not yet ready for publication. However, I see promise in the authors' approach and the potential for this work to make a meaningful contribution to the field. I have aimed to provide constructive feedback that I hope will guide the authors in addressing these concerns and strengthening their manuscript if the authors do agree with them.
>
> With incorporating these suggestions—acknowledging relevant prior work more thoroughly, refining the presentation of their methods, and clarifying key assumptions—I believe this could substantially enhance the impact and rigor of this work. I am open to further discussions if the authors would like to engage with any of the feedback in more detail.

---

> > ### Author Response · Authors · 2024-12-02
> > **Final clarifications**
> >
> > Thanks for your comments. Your suggestions about presentation are well taken. We will incorporate them in the manuscript.
> >
> > **On attribution of Li et al (2024)**
> >
> > Li et al (2024) has not been published in a conference proceeding or a journal. ICLR policy suggests that we are not expected to cite such work (https://iclr.cc/Conferences/2025/FAQ). Nevertheless, we think it’s important to cite it properly, as we did in the revised draft. If the reviewer can suggest specific edits to our citation, we are happy to consider it.
> >
> > **Ambiguous requests**
> >
> > The reviewer made a couple of requests that require clarification.
> >
> > - **Experimental comparison matches Li et al (2024) to the best of our knowledge:** Despite there being no unambiguous description or implementation of FACT’s algorithm in Li et al (2024), we followed the methodology laid out in Fig 5 of Li et al (2024) to the best of our ability. Please point out what specifically was unfair about our comparison.
> > - **Li et al (2024) provides no formal definitions, making formal comparison difficult:**
> > Li et. al 2024 does not provide a formal definition of the bias the method targets, or formal analyses of the proposed methods. Therefore, it is not possible to compare their strategies to OOC in formal terms as the reviewer proposes, e.g. if OOC is an specific case of Li et al (2024). If the reviewer disagrees, they can point us to unambiguous formal definitions/analyses in Li et al 2024 and we will update the paper with a more formal comparison.

---

> > > ### Author Response · Authors · 2024-12-02
> > > **Final clarifications cont'd**
> > >
> > > **Factual errors in the reviewers comments**
> > >
> > > The reviewer made a number of factual errors in their representation of our work:
> > >
> > > - **Our algorithm is not affected by the “internal bias” described the reviewer:** Take their  example of a predictor with racial bias on a data set with a protected attribute in {white, black, latino} that always returns “Yes” for {black, latino}. In this case, the majority vote will *always* return “Yes” for all instances and OOC will satisfy stratified invariance, i.e., OOC succeeds.
> > > - **Our algorithm does not require uniform sampling nor enumeration.** Definition 3 does not require Z+ to be uniform. It can be a constant.

---

> ### Comment · Reviewer_Q4ph · 2024-12-02
> **Further discussions to be done: Feedback IV**
>
> Thank you for your response.  First, I would like to restate that at the current stage, I am not requesting any additional experimental evaluations from the authors. All I am asking is to prevent a **simplified or partial** presentation of the overlap with the very related previous work, and such good practice could also greatly enhance the impact and rigor of this work. Below, I outline the key areas where additional clarity and discussion would be beneficial. These points may also help clarify a few previous misunderstandings:
>
> 1. The statement by the authors that "no unambiguous description or implementation of FACT's algorithm" is **not true**.
> Li et al. (2024) provide detailed theoretical descriptions (e.g., Condition I and Equation (1) under "Strategy I" in Section 3.2 on the left column of page 5), a concrete prompt template (top of the right column of page 5), and exact prompts for each strategy (e.g., WinoBias in the bottom right column of page 6 and Discrim-Eval in Appendix C.2 on page 15).
> After a careful review, Figure 5 is where Li et al. (2024) present the experimental results showing the effectiveness of combining FACT with other prompting strategies, with lighter shades denoting these combinations.
>
> 2. The statement that "FACT requires manual context removal (obfuscation) from inputs" is also **not accurate**. Li et al. (2024) have proposed template(s) automate the generation of base questions, negating the need for manual removal. Such template(s) can be as simple as replacing protected attributes with anaphoric references, streamlining the process.
>
> 3. It is also **unfair** to single out FACT as the only debiasing contribution Li et al. (2024). As stated in my earlier replies, they present three different prompting strategies (Strategies I, II, and III in Section 3.2, supported by Equations 1-4 and specific independence conditions or assumptions). Each prompting strategy has a distinct but related emphasis, and Li et al. (2024) have explicitly claimed that `debiasing is better realized when the strategies are combined as they can address social bias in LLMs more comprehensively`.
>
> 4. For the unambiguous formal definitions/analyses of the three strategies in [1], besides the aforementioned references in Section 3.2, Section 3.1 and Figure 4 provide valuable insights into the causal graphs associated with different data-generating processes. Specifically,
> Section 3.1.1 describes the data-generating process of the training data corpus, while Section 3.1.2 analyzes the potential reasoning process of LLMs. These references should help contextualize Feedback I.3.
>
> 5. After reading through the authors' explanation, I am **more concerned** about how OOC would perform in real-world debiasing tasks, especially under high-stake decision-making contexts. It is crucial to recognize that **there can be many methods satisfying stratified invariance (SI) but not all of them can generate practical or ideal debiasing outcomes.** For example, we can have a constant predictor that always outputs "Yes" for all instances, but is it an effective debiasing method? Consider the example again:
> `Input: "Given a scenario ..., decide whether {a given ethnicity} is a criminal?"; Ground truth: "No"`
> As the author mentioned, OOC would satisfy SI by outputting "Yes" for all ethnicities since the majority vote will always return “Yes” for all instances. Is this really the debiasing result we want? **The goal should be to answer correctly across all demographic groups rather than incorrectly for all groups. Both situations satisfy SI, but the former one should be the one we aim to pursue in the fairness community.** While achieving this is undoubtedly challenging, discussions on trade-offs, such as bias-informativeness, and improvements to OOC. Again, I am not requesting additional experiments, all I am asking is to provide enough discussions that could also help enrich OOC's potential. At a minimum, the authors should address the bias-informativeness trade-off under the implementation of OOC.
>
> 6.   From my understanding, the context obfuscation and addition processes in Algorithm 1 essentially can be used to generate different 'demographic-agnostic fact' and 'demographic-aware text' correspondingly. The effects of these texts could influence the LLM's potential decision via the selection variable "prompt properly considered" (PPC) in Figure 4c) of [1] (e.g., by inputting these pairs and LLMs' answer as in-context examples or by majority voting at the end). As highlighted in Feedback I.3, his process addresses only part of biased causal pathways.
> Therefore, combining OOC with other selection mechanisms to counteract historical biases (i.e., Strategy II and III in [1]), is essential for fair and accurate outcomes across all demographic groups. This can be discussed in future work sections.

---

> > ### Author Response · Authors · 2024-12-02
> > **Addressing reviewer's last concerns**
> >
> > * **Unfortunately, we believe the reviewer’s remaining concern about our work disagrees with existing fairness literature, including previous works published in this venue**. _“The goal should be to answer correctly across all demographic groups rather than incorrectly for all groups”_. The majority of accepted fairness metrics in the literature are unrelated to accuracy, which is why we and others evaluate the effect of enforcing fairness on accuracy empirically ---see in Figure 3 how our method does not impact predictive performance. Although accuracy can be preferred in some situations, it’s also trivial to construct an example where an accurate algorithm doesn’t achieve fairness (e.g. when Y=Z). This is true for classical and established notions of fairness, e.g. demographic parity, and new causal notions, e.g. counterfactual fairness and stratified invariance (ours). Returning a constant output will always define fair predictors according to them.
> >
> > * **The passages in Li et al (2024) that the reviewer is referring to are natural language paragraphs or illustrative notation, and are not, in our reading, clear or rigorous enough to derive a formal comparison with OOC**:
> >
> >      - The reviewer refers to the following in Li et al (2024) for a formal description of the algorithm:
> >        > An example prompt employing Strategy I can be: “Considering the fact that the sentence ‘The physician hired the secretary because the secretary is highly recommended’ is practically more viable than the sentence ‘The physician hired the secretary because the physician is highly recommended’, who does ‘he’ refer to in ‘The physician hired the secretary because he is highly recommended’?
> >
> >        The above is an example prompt for a specific task/dataset and not a formal algorithm or template.
> >
> >     - The reviewer refers to Eq 1 in Li et al (2024) for a formal description of their theory.  The reference is a single conditional independence statement with no formal connection to the method presented.
> >     -  The reviewer claims that the work does not require manual context removal. The suggested automation of context removal requires an in-place substitution as the reviewer mentioned, which is infeasible in real-world tasks with free-form text inputs (we generally do not know where Z will be, or if it’s latent, i.e. we have to manually check).
> >
> >      - Finally, the authors themselves only evaluate FACT in the task discrim-eval, the other prompts in Figure 5 are not from them.
> >
> > **We restate that, despite us not being required by ICLR’s policy to discuss Li et al (2024)’s draft, we used ~50% of a related work paragraph to discuss it, added empirical results comparing the same prompt used in the original work to our method, and we are happy to consider any specific changes in the citation the reviewer suggests.**

---

> ### Comment · Reviewer_Q4ph · 2024-12-02
> **Further Discussions: Feedback V**
>
> Thanks for the authors' response!
>
> 1. The performance trade-off is a common topic of discussion in many existing works on fairness and safety (e.g., [1]), as methods aimed at enhancing fairness and safety often result in reduced performance. It is important to **analyze or at least acknowledge** this trade-off. The performance drop of OOC observed with standard prompting in Figure 7 on the Bios and Clinical datasets also provides evidence of this phenomenon.
>
> 2. If the OOC strategy does not rely on knowledge of $Z$, could the authors clarify what information is used in Prompt 13 and Prompt 14 to populate the fields {Z_list}, {Z_description}, and {random_Z}?
>
> 3. If the OOC algorithm does not depend on uniform sampling or enumeration, Line 4 of Algorithm 1 should be revised. As written, it samples $Z_+^j$ uniformly from $\mathcal{Z}$, which seems inconsistent with what is stated by the authors.
>
> 4. If time allows, I also look forward to discussing the issues raised in Feedback I.2 (i.e., conceptual alignment between context obfuscation/addition and the 'demographic-agnostic fact'/'demographic-aware text' in [2]), III.1, III.2, III.3 (presentation improvements), and IV.3 with the authors.
>
> [1] Parrish, Alicia, et al. "BBQ: A hand-built bias benchmark for question answering." arXiv preprint arXiv:2110.08193 (2021).
>
> [2] Li, Jingling, et al. "Steering LLMs Towards Unbiased Responses: A Causality-Guided Debiasing Framework." arXiv preprint arXiv:2403.08743 (2024).

---

> > ### Author Response · Authors · 2024-12-02
> > **Final clarifications**
> >
> > Thank you for clarifying your doubts.
> >
> > (1.) We have an entire subsection in the experiments, starting in ln 478 not only recognizing the need to report performance tradeoffs, but also reporting it for every prompt. This is the first sentence of the subsection:
> > > Stratified invariance does not guarantee strong predictive performance.
> >
> > (2. and 3.) As we state in the paper, and as the reviewer commented already, OOC is an implementation of Theorem 1, i.e. it instantiates it. We chose to enumerate Z in the prompt template and do uniform sampling to provide the reader with a concrete implementation of a general-purpose, flexible algorithm. Changing the sampling or the prompt template is always a (trivial) possibility in any task.
> >
> > (4.) Regarding the discussion surrounding Li et al (2024), we have already extensively clarified how the original work i) uses the same prompt as we did in our comparison, and ii) does not provide theoretical/formal results that would allow us to provide a deeper connection between the methods. The reviewer points to comments already answered by us, but if they are more specific about what's unresolved we can try to clarify more concrete questions.
> >
> > **Finally, ICLR's policy is very clear about the extent to which unpublished drafts have to be addressed. The reviewer had initially committed to raising their scores based on new experiments, so despite not being required to do so, we cited, empirically compared, extensively discussed in a public thread, and are willing to adapt the citation in a way the reviewer concretely proposes.** As we approach the end of the discussion period, we thank you for the service once again!

---

### Author Response · Authors · 2024-11-18
**General Response**

We thank the reviewers for their thorough feedback. We are particularly encouraged that you found our work's theoretical and empirical contributions "interesting, original, of clear presentation, and of superior performance." We have carefully addressed all questions and suggestions, including adding new experimental results that further strengthen our claims.

**All updates to the manuscript are highlighted in blue for easy reference.**

Below, we first address a key point raised by two reviewers, followed by detailed responses to specific questions/suggestions in each reviewer thread.

### Comparison with Li et al. (2024)

We address reviewers Q4ph and 86wE's questions about Li et al.'s recent pre-print (FACT). While both works address LLM debiasing at test time, there are fundamental differences in approach and applicability:

1. **Causal Framework:**
- **FACT** addresses only selection bias through $Z$.
- **OOC** handles broader bias patterns through stratified invariance, including selection bias.

2. **Algorithmic Design:**

- Unlike OOC, **FACT** requires manual context removal (obfuscation) from inputs.
- FACT still uses the original (**not obfuscated**) input to make the final prediction.

**Empirical Comparison:** We evaluated FACT against OOC on the synthetic discrimination task from Section 5.1, the only setting where FACT's manual context removal is feasible. Results in Tables 1-3 (Appendix E) show:

- OOC outperforms FACT in 8/9 of our settings.
- Default prompting performs at least as well as FACT in 6/9 of our settings. Indeed, FACT also does not provide considerable gains over default prompting in Li et al.’s implementation of this task.
- This underperformance may stem from FACT being shown the original question, allowing the model to rely on information in the protected attribute Z.

**Updates to Manuscript:**

- Added FACT discussion to related work.
- Referenced comparison in Section 5.1.
- Included raw results numbers in Appendix E (highlighted in blue).

---

### Author Response · Authors · 2024-11-28
**Extension**

Dear reviewers,

Following our rebuttal submission 10 days ago, which included comprehensive responses to your feedback and the requested additional experimental results, we haven't yet received your updated evaluations.
Given that we have 6 days remaining in the extended review period, we would greatly appreciate your feedback soon to allow time for any necessary clarifications or adjustments. Your insights are vital to improving our manuscript through a proper peer review process.

---

### Comment · Senior_Area_Chairs · 2024-11-30
**Please provide your response to the authors' rebuttal ASAP**

Dear Reviewers,

*Would you please respond to the authors' rebuttal ASAP?* We are drawing close to the end of the author-reviewer discussion.

Many thanks for your reviewing effort!

Your SAC

---

### Author Response · Authors · 2024-12-03
**Wrapping up the discussion period**

Dear AC/SAC, PC, and Reviewers,

Thank you for your thoughtful feedback during the discussion period. Your comments have helped us improve the presentation of our work. We believe we have addressed the concerns, resulting in improved review scores. We appreciate reviewers Q4ph and 86wE bringing attention to the recent pre-print of Li et al (2024), which we have now incorporated through both citation and empirical comparison.
We appreciate your time and attention in reviewing our submission.

---

### Meta-Review · Area_Chair_CDux · 2024-12-21

**Metareview:**

This paper uses causal inference to achieve test-time fairness and robustness of LLMs. The paper proposes stratified invariance which provides a better measurement of biases of LLM-generated texts, and proposes a strategy to achieve stratified invariance.

The reviewers in general agree that the theoretical analysis and results are interesting and useful, and that the experiments are well designed and the results are promising.

During the rebuttal period, there were extensive discussions regarding comparisons with the related work of Li et al. (2024). In particular, both Reviewers Q4ph and 86wE pointed out this very related work and mentioned the similarities between the proposed methods from Li et al. (2024) and the current paper. For example, Reviewer Q4ph commented that there is a "conceptual alignment between context obfuscation/addition (from this work) and the 'demographic-agnostic fact'/'demographic-aware text' (from Li et al. (2024))". From what I have read in the discussions, this point was not explicitly clarified or acknowledged by the authors. For the healthy development of a field, I think it is important to clearly position the contribution of a work with respect to the prior works and provide fair and accurate descriptions and attributions to the contributions from the prior works. In addition, there are other concerns shared by multiple reviewers as well, for example, the clarity of presentation can be improved.

Given these concerns and also given that no reviewer would like to champion the paper, rejection is recommended.

**Additional Comments On Reviewer Discussion:**

During rebuttal, there were extensive discussions about the comparison of the method from this work and the previous work of Li et al. (2024). After the discussion, the conclusion is that a more accurate and comprehensive discussion of this related work is needed to better position the current paper in the field.

---

### Decision · Program_Chairs · 2025-01-22

Reject